# Efficient Subgraph Rule Induction via Tree Folding in Differentiable Logic Programming

## Abstract

Differentiable inductive logic programming techniques have proven effective at learning logic rules from noisy datasets; however, existing algorithms incur pernicious trade-offs between rule expressivity and scalability to large problems. Forward-chaining ILP algorithms can learn arbitrary rules, but their memory requirements scale exponentially with problem size. Backwards-chaining ILP algorithms address this limitation, but do so with loss of generality by imposing the restrictive constraint that rules must be expressible as ensembles of independent chain-like Horn clauses. In this paper we present Folding and Unification for Subgraph Expression in Inductive Logic Programming (FUSE-ILP), a technique that significantly extends the expressive capability of ILP, enabling the differentiable evaluation of a rich class of subgraph-like rules. Our method extends TensorLog-inspired backwards-chaining ILP techniques with the introduction of a tree-like clause template and a differentiable set partitioning technique for leaf grouping. Together these enable tree-like rule evaluation, and the "folding" of these trees into complex subgraphs. We demonstrate that this formulation allows our algorithm to learn more expressive rules than previous backwards-chaining algorithms while retaining a similar computational cost.

## 1 Introduction

Inductive logic programming (ILP) is an interpretable machine learning technique that seeks to learn highly generalizable models from low amounts of data. The models produced by ILP are expressible as first-order logic rules. These rules can be human interpretable and reveal information about the underlying structure of the data and relationships therein. While the ability of ILP to learn patterns from few examples is powerful, it is often desirable to learn ILP-style rules in the context of large problems for which current ILP algorithms are unsuitable. Real-world problems in areas such as financial crime, fraud, molecular biology, and cybersecurity all produce massive graph datasets which cannot be processed by typical ILP techniques.

Historically, ILP techniques have used custom symbolic solvers written in logic programming languages such as Prolog (Muggleton, 1995; Shrinivasan, 2001; Muggleton et al., 2015). These solvers use hypothesis generation algorithms such as inverse entailment alongside heuristic search and meta-templating strategies to efficiently find theories which entail the positive examples in a dataset while rejecting the negative examples. More recent techniques such as Cropper & Morel (2021) map the ILP problem into an answer set programming (ASP), or Boolean satisfiability (SAT) problem, which can be solved with existing high-performance SAT solvers.

Recent research has developed a class of ILP solver which implements a restricted version of the ILP problem using end-to-end gradient-based optimization. Typically, these techniques are trained and evaluated as graph link-prediction models and learn rules that classify relationships in a graph. Differentiable ILP systems accomplish this by numerically implementing approximate rule inference. This is the process of proving a rule true or false for a specific sample from some background data. These techniques numerically parameterize a search space of possible rules and, given a training dataset, optimize through the inference process to learn which rule in the search space correctly classifies the most positive and negative examples.

The numerical nature of differentiable ILP systems makes many features common to symbolically implemented ILP systems, such as datatypes and arithmetic operations, challenging to reproduce.

Despite these limitations, there are several features that make differentiable ILP more suitable than symbolically implemented ILP on certain tasks. The most significant advantage of differentiable ILP techniques over symbolic ILP techniques is their ability to handle large problem sizes. YAGO3-10, a common benchmark dataset for link prediction and information retrieval tasks has nearly 1.2M triples, making it far too large for ILP systems that encode the entire background knowledge into a SAT or ASP program. Differentiable ILP techniques have been shown to perform well when applied to such large graphs, and they can be implemented efficiently using sparse matrix operations on GPUs (Yang et al., 2017; Yang & Song, 2020). Differentiable ILP techniques are naturally parallel and typically support iterative batch optimization, further extending their ability to scale to large problems and large compute resources. Finally, differentiable ILP techniques have been shown to offer robustness to noise, a feature that some symbolically implemented ILP techniques such as Metagol and Popper[1] lack (Evans & Grefenstette, 2018; Cropper et al., 2022).

Previous backwards-chaining differentiable ILP techniques such as Yang et al. (2017); Yang & Song (2020) have imposed syntactic restrictions on the rules that can be learned, principally a chain-like rule restriction (Cropper et al., 2022). These methods have performed well in ontological challenges such as information retrieval on encyclopedic data sources and classification of kinship terms, however their structural limitations render many real-world patterns un-learnable. In this paper we introduce FUSE (Folding and Unification for Subgraph Expression) ILP, a technique for expressing and learning more syntactically expressive rules.

## 1.1 MOTIVATING EXAMPLE: ANTI-MONEY LAUNDERING

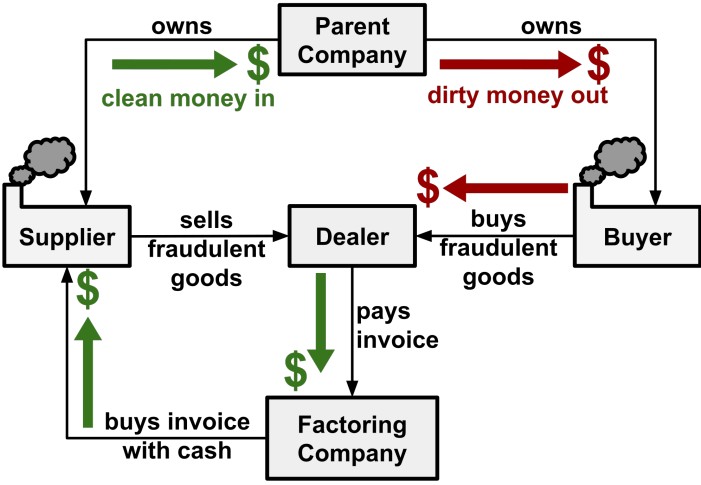

Figure 1: Money-laundering network illustrating non-chain-like patterns.

To illustrate the importance of non-chain-like reasoning, consider the money laundering network depicted in Figure 1. Suppose a parent company has illegitimately obtained money and wishes to replace it with legitimate money. This parent company tasks two of its subsidiaries with carrying out fraudulent transactions though an intermediary. A supplier sells a palette of fake GPUs to a third-party dealer and issues an invoice for the dealer to pay at a future date. The supplier then sells this invoice to a factoring[2] company for legitimate money. The dealer auctions off its newly acquired palette of fake GPUs, and the colluding buyer, knowing they are fraudulent, out-bids the other buyers for the GPUs. The buyer pays the dealer for the GPUs with illegitimate money, and the

---

[1]Wahlig (2021); Hocquette et al. (2023) have since introduced extensions of Popper that enable rule learning from noisy examples.

[2]A factoring company purchases accounts receivables from businesses at a discount and then collects payments from the customers directly. This allows businesses to receive immediate cash for invoices, rather than waiting for their customers to pay. Factoring companies are typically subject to regulations designed to prevent money laundering.

dealer settles the invoice with the factoring company that now owns it. In this process, the parent company replaces illegitimate money with legitimate money at some loss.

In this example, the supplier and buyer are involved in a money laundering scheme which we can inductively generalize into a rule. Using first-order logic, we construct a rule which states that any time this pattern of interactions occurs between two entities, they are likely to be involved in money laundering.

$$\forall X, Y (\text{Money Laundering}(X, Y) \leftarrow \exists Z_1, Z_2, Z_3 (\text{Owns}(Z_1, X), \text{Owns}(Z_1, Y),$$
$$\text{Sells}(X, Z_2), \text{Sells}(Z_2, Y),$$
$$\text{Buys}(Z_3, X), \text{Pays}(Z_2, Z_3))).$$

In the first-order logic expression above, we use universally quantified variables $X$ and $Y$ to assert that the rule should hold true for any possible pairing of entities that could be substituted into these variables. The free variables, $Z_1, Z_2, Z_3$ are existentially quantified, meaning that if there exists at least one substitution of entities into these variables which satisfies the conditions in the rule body for a specific $X, Y$, then those $X$ and $Y$ entities are involved in money laundering. The $\leftarrow$ indicates that the truth of the expression on the right implies the truth of the expression on the left.

To illustrate how this rule applies to the previous example, we construct a substitution using the entities in Figure 1. Each entity is identified by its initials; $S$ is the seller, $B$ is the buyer, $D$ is the dealer, $PC$ is the parent company, and $FC$ is the factoring company.

$$\text{Money Laundering}(S, B) \leftarrow \text{Owns}(PC, S), \text{Owns}(PC, B),$$
$$\text{Sells}(S, D), \text{Sells}(D, B),$$
$$\text{Buys}(FC, S), \text{Pays}(D, FC).$$

Each *Interaction*(*Source Entity*, *Target Entity*) triple in the rule is called an atom and describes a Boolean condition on the relationship between its argument entities. For this specific substitution of entities, the relationships described by each atom in the body all exist in the Figure 1 graph. This implies that the supplier and buyer are colluding to launder money according to the rule.

Critically, the generalized rule describing this money laundering relationship is non-chain-like, meaning that it cannot be factored into independent clauses of the general form $P_1(X, Z_1), P_2(Z_1, Z_2), ..., P_n(Z_{n-1}, Y)$. This renders it impossible to represent in differentiable ILP frameworks which learn collections of chain-like rules such as Yang et al. (2017); Yang & Song (2020). Many real-world problem domains contain data which exhibit graph-like patterns such as the one demonstrated here. To enhance the applicability of differentiable ILP to these domains, FUSE-ILP extends previous works by introducing mechanisms to enable graph-like rule learning.

## 2 RELATED WORK

Many neural, symbolic, and hybrid techniques have been introduced for the fundamental task of generalizing from specific training examples to symbolic rules that explain the examples. The framework of inductive logic programming (ILP) looks to represent such explanations as logic programs (Cropper et al., 2022). ILP falls under the broader category of statistical relational learning, which include expert systems, Relational Markov Networks, and Markov Logic Networks (Richardson & Domingos, 2006). Other representations, such as Graph Neural Networks (Kipf & Welling, 2017) have also demonstrated effectiveness (Battaglia et al., 2018) at this task, including in the generation of first order logic formula. Embedding based methods provide state-of-the-art in performance, although unsatisfactory due to their lack of interpretability. ILP based methods have demonstrated performance on par with embedding methods in the complex task of link and node prediction in temporal knowledge graphs (Xiong et al., 2023), with a simplifying assumption on the rule structures.

The concept of representing knowledge bases in a differentiable semantically meaningful representation was introduced as TensorLog in Cohen et al. (2020), and extended to inductive logic learning of chain-like rules of knowledge structures in Yang et al. (2017). The first morphological extension of this approach joined chain-like rules at a single node (Yang & Song, 2020) by reducing such a structure to a chain-like rule constructed from the original differentiable predicates and their adjoints. Allowing joining of rules extended the experessivity and was later shown to be sufficiently

general and computationally efficient for application to countering adversarial machine learning attacks using real-world-scale commonsense rule learning and reasoning (Yang et al., 2022).

Powerful differentiable forward-chaining first-order-logic methods have been demonstrated in Evans & Grefenstette (2018), where soft truth tables were employed to iteratively and differentiably derive the consequences of all facts in a database. $\delta$ILP was the first neural approach to ILP that implemented traditional features of ILP such as predicate invention and recursion together. A more computationally efficient method using differentiable normal-form representations was developed in Payani & Fekri (2019).

# 3 BACKGROUND

## 3.1 DIFFERENTIABLE INDUCTIVE LOGIC PROGRAMMING

Inductive logic programming is a machine learning technique that attempts to inductively learn logical hypotheses from observations. These hypotheses are typically described using first-order logic, a formal system that extends propositional logic with quantified variables and predicates, as demonstrated in the money-laundering example.

In FUSE-ILP, the inductive logic programming task is interpreted as a supervised binary link-prediction problem on graph data. A dataset of (source entity, relation, target entity) triples captures the background knowledge for the learning task. These triples are interpreted as a directed graph of relationships between the entities in the background domain. An additional set of triples encodes the supervised examples that a model will learn to classify. This set contains properties of (in the case of unary predicates), or relations between (in the case of binary predicates), entities in the background knowledge graph. FUSE-ILP can learn a single type of relationship at a time, so the examples set will contain a series of triples labeled as positive or negative examples of the "target" relationship or property.

In differentiable ILP methods inspired by TensorLog, a differentiable database, background data is numerically encoded into a collection of binary graph adjacency matrices representing the connectivity of each edge type in the graph (Cohen et al., 2020; Yang et al., 2017; Yang & Song, 2020). Entities are numerically encoded as one-hot (for a single entity) or many-hot (for multiple entities) vectors. Backwards-chaining differentiable ILP models implement a continuous relaxation of logical rule inference. The model ingests the numerically encoded background graph and vectors representing the indices of the entities involved in the queried edge. The model returns a probability that the modeled edge exists between the query entities. Gradient-based optimization is used to fit this model to the training examples, and an explicit logical rule can be extracted from the model after convergence. In the case of FUSE-ILP, learned rules are syntactically expressive, including cycles and "dangling" existentially quantified variables. This enhanced expressivity is the primary contribution of FUSE-ILP over previous differentiable ILP methods which could only represent chain-like rule syntax.

## 3.2 FORMULATING ILP AS GRAPH STRUCTURE LEARNING

Let $(x, y, \mathcal{G}) \in \mathbb{D}$ be a triple of head entity $x$, tail entity $y$, and local subgraph $\mathcal{G}$ drawn from a dataset $\mathbb{D}$. $\mathcal{G}$ is a typed, directed multigraph with edge types drawn from a set of first-order-logic predicates $\mathbb{P}_{\mathcal{G}}$ and entities drawn from $\mathbb{E}_{\mathcal{G}}$. $x, y \in \mathbb{E}_{\mathcal{G}}$ form a positive pairing for some predicate $p \in \mathbb{P}_{\mathcal{G}}$ if there exists a directed edge of type $p$ from $x$ to $y$ in $\mathcal{G}$.

For any edge type $p \in \mathbb{P}_{\mathcal{G}}$, it is convenient to define a child operator $\phi_p : \mathbb{E}_{\mathcal{G}} \mapsto \mathbb{E}_{\mathcal{G}}$ that maps a subset of the graph's entities $\mathbb{S} \subseteq \mathbb{E}_{\mathcal{G}}$ to another subset of the graph's entities[3] $\mathbb{T} \subseteq \mathbb{E}_{\mathcal{G}}$, such that $\mathbb{T}$ contains the union of the sets of child entities by relation type $p$ of the entities in $\mathbb{S}$.

$$\phi_p(\mathbb{S}) = \bigcup_{s \in \mathbb{S}} Ch_{\mathcal{G},p}(s) \tag{1}$$

Where $Ch_{\mathcal{G},p}(s)$ is the set containing all entities in $\mathcal{G}$ that have an incoming edge of type $p$ from the entity $s$. A parent operator can be defined in a similar manner, mapping a subset of graph entities $\mathbb{S}$

---

[3]$\mathbb{S}, \mathbb{T}$ are chosen to indicate a "source" set and "target" set.

to the union of the parent sets of those entities by edge type $p$.

$$\phi_{p^{-1}}(\mathbb{S}) = \bigcup_{s \in \mathbb{S}} Pa_{\mathcal{G},p}(s) \tag{2}$$

Where $Pa_{\mathcal{G},p}(s)$ is the set containing all entities in $\mathcal{G}$ that have an outgoing edge of type $p$ into the entity $s$. The parent operator for some predicate $p \in \mathbb{P}_{\mathcal{G}}$ is equivalent to the child operator for the reversed relationship $p^{-1}$ (wherein all directed edges of type $p$ in $\mathcal{G}$ are re-mapped to point in the opposite direction). However, these operators are not true inverses, as neither is injective. Real-valued implementations of these operators using matrix multiplication will become the backbone of the message passing algorithm which models rule inference.

## 4    LEARNING SUBGRAPH RULES

In FUSE-ILP, subgraph-like rules are learned by initializing a model with a tree-like rule structure and performing simultaneous bi-level iterative optimization to transform the initial tree into one of many reachable rules. Structural optimization is performed in three ways. Branches from the meta-rule tree are stretched or contracted to include chains of multiple predicates, sub-trees are softly pruned from the tree, and leaf variables are softly merged to form cycles. Non-structural optimization of the rule is performed by softly selecting different predicates for each edge in the rule graph. These high-level strategies are depicted in Figure 2.

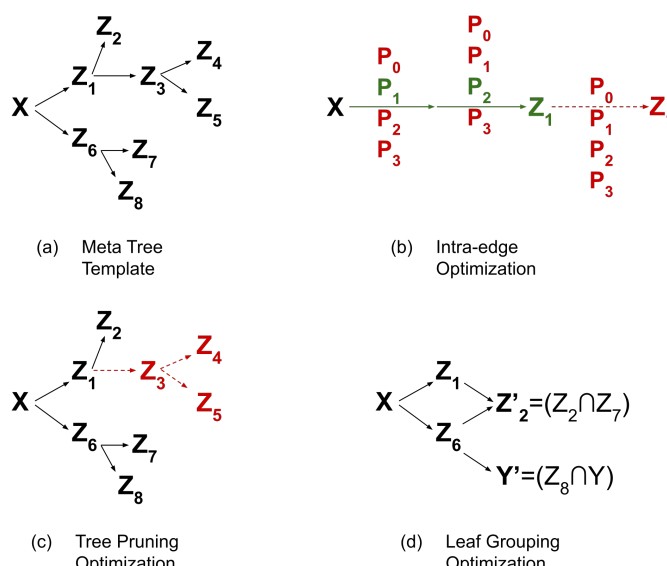

(a)    Meta Tree Template

(b)    Intra-edge Optimization

(c)    Tree Pruning Optimization

(d)    Leaf Grouping Optimization

Figure 2: Illustrations of the different stages of optimization that FUSE uses to capture complex rule structures. (a) An example of a tree template parameterizing the rule search space. (b) FUSE learns variable-length chains of predicate relationships along each edge in the tree template. (c) FUSE includes a pruning group in its leaf partition search, and sub-trees with pruned leaves are softly eliminated from tree message passing. (d) FUSE learns a leaf partition set which combines existentially quantified leaf variables and assigns head variables to leaves. This enables FUSE to learn cyclic rules.

### 4.1    GENERAL RULE EVALUATION

When existentially quantified variables are included in a rule body, evaluating the truth of the rule requires the inference engine to prove or disprove the existence of a set of entities in the background graph that could be substituted into the variables and produce a true clause. The process of proving

this for a given inference can be expressed as a Boolean satisfiability (SAT) problem. As described by Cohen et al. (2020), general inference of a FOL clause on a graph can be accomplished using a message passing belief propagation algorithm on the factor-graph formed by the logic variables of the clause and the atoms joining them together. This paper will use a simplified formulation of Cohen et al. (2020)'s message passing algorithm, to prove that clauses are satisfiable when applied to some head entities in a graph.

For a tree-like FOL clause applied to some head entities in $\mathcal{G}$, let the *rule graph* $\mathcal{R}$ be a directed factor graph where the set of nodes $\mathbb{N}_\mathcal{R}$ represent the logic variables of the rule, and the set of factors between them $\mathbb{P}_\mathcal{R}$ represent the atoms involving each variable. To solve the satisfiability problem for a rule with graph $\mathcal{R}$ applied to $\mathcal{G}$, a message passing algorithm over the graph is employed. Message passing takes a triple $(\mathcal{R}, x \in \mathbb{E}_\mathcal{G}, \mathcal{G})$ and determines whether there exists an instantiation of the logic variables in the rule represented by $\mathcal{R}$ such that every atom of the rule body evaluates to true when the head variable $X$ is instantiated as the entity $x$ in $\mathbb{E}_\mathcal{G}$. Due to the underlying tree structure of the rule graph in FUSE-ILP, this algorithm is guaranteed to resolve the satisfiability of the rule after a forward and backward pass through $\mathcal{R}$[4].

## 4.2 MESSAGE PASSING ON RULE FACTOR GRAPHS

During message passing, each node in the factor graph $Z_k$ will have an associated set $\mathbb{Z}_k^{(i)}$ representing the $i$th estimate of that variable's satisfying set of entities from $\mathbb{E}_\mathcal{G}$. Message passing will begin with the root node $Z_r$ and instantiate its SAT set as $\mathbb{Z}_r^{(0)} = \{x\}$. During the forward pass, every variable with a defined $\mathbb{Z}_k^{(0)}$ will send a message $\psi_k^{(0)}$ along its outgoing edges[5].

$$\psi_k^{(0)} = \mathbb{Z}_k^{(0)} \tag{3}$$

Each factor will receive a single incoming message from its parent node and update the message by applying the child or parent operators from equation 1 or equation 2 associated with its modeled atom. It will then pass the updated message to its child node.

$$\psi_k'^{(0)} = f_p(\psi_k^{(0)}) \tag{4}$$

Generally, each non-root node in the factor graph will receive a set of $N$ incoming messages from its parent factors $\{\psi_j'^{(0)}\}_{j=1}^N$ and define its SAT set estimate $\mathbb{Z}_k^{(0)}$ to be the intersection of these incoming messages.

$$\mathbb{Z}_k^{(0)} = \bigcap_{i=1}^N \psi_j'^{(0)} \tag{5}$$

For tree-like rules, every node has an in-degree of 1, so equation 5 is equivalent to $\mathbb{Z}_k^{(0)} = \psi_1'^{(0)}$. This condition will be invalidated for non-tree rule structures and equation 5 must be applied in its general form.

Once messages have propagated from the root node to every leaf node, and each variable in the graph has a defined $\mathbb{Z}_k^{(0)}$, the backward pass can begin. All leaf nodes initialize an updated SAT set $\mathbb{Z}_k^{(1)} = \mathbb{Z}_k^{(0)}$. During the backward pass, every variable with a defined $\mathbb{Z}_k^{(1)}$ will send a message $\psi_k^{(1)}$ back up the tree along its incoming edges to all its parent factors.

$$\psi_k^{(1)} = \mathbb{Z}_k^{(1)} \tag{6}$$

Every factor will receive an incoming message $\psi_k^{(1)}$, generate an updated message $\psi_k'^{(1)}$, and forward it up the tree to its parent node. Updated factors are calculated by applying the predicate inverse operator of that applied by the factor during the forward pass.

$$\psi_k'^{(1)} = f_{p^{-1}}(\psi_k^{(1)}) \tag{7}$$

---

[4]While the satisfiability of a tree-like rule can be verified in two passes, full resolution of the SAT sets (specific entity substitutions entailing clause truth) for each variable would require additional message passing.

[5]Note that all edges in the factor graph point away from the root. Without loss of generality, the atoms associated with these factors can be replaced by equivalent atoms of inverse predicates to maintain this convention. For example, $P(Z_1, X) \mapsto P^{-1}(X, Z_1)$, $(\mathbb{X} = f_p(\mathbb{Z}_1)) \mapsto (\mathbb{Z}_1 = f_{p^{-1}}(\mathbb{X}))$.

Each non-leaf node in the factor-graph will receive an incoming message from each of its outgoing edges. A node with out-degree $N$ will receive $N$ incoming messages from its child factors. Each node will aggregate its received messages, $\{\psi_j'^{(1)}\}_{j=1}^N$, and generate an updated SAT set $\mathbb{Z}_k^{(1)}$ from the intersection of the previous SAT estimate $\mathbb{Z}_k^{(0)}$ and the incoming messages.

$$\mathbb{Z}_k^{(1)} = \mathbb{Z}_k^{(0)} \bigcap_{j=1}^N \psi_j'^{(1)} \tag{8}$$

When the $\mathbb{Z}_k^{(1)}$ associated with every node in the factor graph is derived, message passing is complete. If the root node's SAT set $\mathbb{Z}_r^{(1)} \neq \emptyset$, then the rule is satisfiable for $x, \mathcal{G}$. In-practice it is sufficient to terminate message passing when the top-most branching variable's $\mathbb{Z}_k^{(1)}$ has been derived. If this $\mathbb{Z}_k^{(1)}$ is non-empty, then the rule is satisfiable.

### 4.3 Folding Trees into Subgraphs

Formally, for a tree-like rule with rule graph $\mathcal{R}$ and leaf variables $\mathbb{N}_{\mathcal{R},L} = \{Z_k\}_{k=1}^L$, we can define a partition of $\mathbb{N}_{\mathcal{R},L}$, $\{\{\mathbb{S}_1\}, \{\mathbb{S}_2\}, ... \{\mathbb{S}_n\}\}$ such that no subset is empty $\mathbb{S}_i \neq \emptyset$, no subsets intersect $\mathbb{S}_i \cap \mathbb{S}_{j \neq i} = \emptyset$, and the subsets completely cover the set of leaf variables $\mathbb{N}_{\mathcal{R},L} = \bigcup_{i=1}^n \mathbb{S}_i$. If an $\mathbb{S}_i$ contains multiple leaf variables, then these leaf variables are "merged" and their corresponding nodes in the factor graph will be evaluated as a single node during message passing.

After the forward pass of the message passing algorithm, initial SAT sets for each leaf node have been derived, $\{\mathbb{Z}_k^{(0)}\}_{k=1}^L$, independent of any partition assignments. If a subset $\mathbb{S}_n$ of these leaf nodes are to be merged, then the intersections of the SAT sets $\mathbb{Z}_k^{(0)}$ associated with each leaf node in the partition will be calculated.

$$\mathbb{Z}_{\mathbb{S}_n} = \bigcap_{\mathbb{Z}_k \in \mathbb{S}_n} \mathbb{Z}_k^{(0)} \tag{9}$$

The running intersection of each set of partitioned leaf nodes, $\mathbb{Z}_{\mathbb{S}_n}$, will replace each individual leaf node's SAT set before the beginning of the backwards pass of message passing.

$$\mathbb{Z}_k'^{(0)} := \mathbb{Z}_{\mathbb{S}_n} \forall k \in \mathbb{S}_n \tag{10}$$

It can be observed that this process is identical to applying equation 5 for the case that partitioned leaf nodes have in-degrees greater than 1, as each merge produces a single variable with multiple incoming edges from parent factors. When implemented differentiably, this operation will perform a soft intersection, capturing a superposition of possible leaf node partitions.

The restriction that variable merges only occur at the leaves of the factor graph ensures that the factor graph remains acyclic and tree-like for the purpose of inference. As a consequence, the previously defined message passing algorithm can, by construction, be used to derive the satisfiability of subgraph-like rules in two fixed passes through the graph.

### 4.4 Differentiable Leaf Partitioning

FUSE-ILP implements differentiable leaf partitioning by learning a parameterized discrete probability distribution over the space of possible leaf set partitions. During message passing, the expected leaf SAT sets $\{\mathbb{Z}_l^{(1)}\}_{l=1}^L$ are computed with respect to this distribution.

To define a probability distribution over possible leaf partitions, a list of indices representing each leaf node is first collected. Additional indices are appended to this list for each head variable, as well as a "prune" index.

$$l = [1, 2, ..., L, x, y, \bot]$$

The set of all possible set partitions of these elements is then calculated. To ensure valid partitions and reduce the number of possible partitions, several simplifying constraints are introduced. Head variables cannot be grouped together, and neither head variable can be assigned to a set by itself. Additionally, if the target rule is binary then the tail variable, $y$, cannot be assigned to the pruning set.

Any partitions violating these constraints are removed, leaving a set of $K$ possible valid partitions. A group partition matrix, $\boldsymbol{G}$ is then constructed for each valid partition. Each group partition matrix is constructed to be row-wise stochastic and one-hot, such that each column represents a group of merged variables and each row represents a variable's group membership. All group partition matrices are stacked to form a new group partition tensor $\mathbf{G}$.

$$\mathbf{G} = [\boldsymbol{G}_{:,:,k}]_{k=1}^K \tag{11}$$

As with factor predicate selection, a parameterized probability mass function is generated at runtime, $f_k(k; \theta)$, and this distribution is used to compute the expected partition matrix.

$$\mathbb{E}_k[\boldsymbol{G}_{\mathcal{R}}] = \sum_{k=1}^K f_k(k; \theta)\mathbf{G}_{:,:,k} \tag{12}$$

This soft partition matrix is then used to compute updated SAT sets for each group. For each group, the corresponding column of the expected partition matrix is indexed. The rows of this vector represent the expected involvement of each leaf and head variable $\mathbb{L} = [\boldsymbol{z}_1, \boldsymbol{z}_2, ..., \boldsymbol{x}, \boldsymbol{y}]$ in the group. The soft SAT sets for each variable in $\mathbb{L}$ are first weighted by their corresponding entry in this vector. These weighted values are then added to the complement of the weight. This has the effect of "turning off" the contribution of non-participating leaf vectors towards the element-wise minimum across the vectors in a group. Within each group, the element-wise minimum is calculated across the weighted vectors to produce a soft SAT set for the variables participating in each group.

$$\mathbb{E}_k[\mathbb{Z}_g] \approx \mathbb{E}_k[\boldsymbol{z}_g] = \min_{l=1,...,|\mathbb{L}|} \left[ \mathbb{E}_k[\boldsymbol{G}_{\mathcal{R}}]_{l,g}\boldsymbol{z}_l + (1 - \mathbb{E}_k[\boldsymbol{G}_{\mathcal{R}}]_{l,g}) \right] \tag{13}$$

Note that the prune column is explicitly ignored to approximate the pruning of non-participating vectors. With each merged group's soft SAT set $\mathbb{E}_k[\boldsymbol{z}_g]$ calculated, each leaf variable defines its new soft SAT set $\boldsymbol{z}_k^{(1)}$ as a weighted combination of the group vectors according to the variable's participation in each group.

$$\mathbb{Z}_l^{(1)} \approx \boldsymbol{z}_l^{(1)} = \sum_{i=1}^{|\mathbb{L}|} \mathbb{E}_k[\boldsymbol{z}_i]\mathbb{E}_k[\boldsymbol{G}_{\mathcal{R}}]_{l,i} \tag{14}$$

The updated soft SAT sets $\mathbb{Z}_l^{(1)}$ for each leaf variable are then substituted in to initialize the backwards pass of message passing.

### 4.5 PARAMETERIZING THE JOINT DISTRIBUTION

FUSE-ILP enables a wide variety of rule structures by making soft approximations of discrete, structural selections at every stage of rule evaluation. These selections are highly inter-dependent, and generally the joint distribution of these structural variables is unknown. FUSE-ILP addresses this problem by introducing a neural network which approximates a joint distribution on the variables defining discrete rule selection. In our experiments, we employ a UniMP graph transformer (Shi et al., 2021) which is initialized with dummy embedding vectors for each node in the meta rule template. At inference time, these embedding vectors are "mixed" by the transformer to produce updated representations which are then projected by shared linear projection layers into the decision dimensions for the various soft selections performed during rule learning. The softmax function is used to create discrete distributions for approximating expectations during soft inference. This component of FUSE-ILP serves as an optimization device, enforcing dependence between decisions and projecting the optimization problem into high dimensions.

## 5 EXPERIMENTS

### 5.1 EXPERIMENTAL SETUP

We compare FUSE-ILP to NLIL (Yang & Song, 2020), a similar ILP technique which builds on Yang et al. (2017) by enabling mixed-directional chains and Boolean combinations of chains, including disjunction. We compare the algorithms on three benchmark tasks. The Kinship dataset (Hinton,

| Model | Kinship | | ES-1k | | Community | |
|---|---|---|---|---|---|---|
| | F1 | Time | F1 | Time | F1 | Time |
| NLIL | - | - | 1.0 | 6 | 0.0 | 30 |
| FUSE-ILP | 1.0 | 4 | 1.0 | 1 | **0.8** | 180 |

Table 1: F1-score and time (seconds) to learn a single predicate on several ILP tasks.

1990) evaluates the models' ability to learn simple, chain-like rules from few examples. The even-successor task, tests the models' ability to learn recursive rules by discovering recursive definitions for even numbers. We introduce the Community dataset to evaluate the ability of each algorithm to learn challenging non-chain-like rules from a small number of examples, similar to the Kinship dataset. This dataset models a community with 14 people: 8 parents and 6 children, their friendships, the schools or football clubs that they attend, and who has met whom. There are four positive examples of a friendship between parents that the models must generalize a rule from.

## 5.2 EXPERIMENTAL RESULTS

Table 1 shows the performance of FUSE-ILP and NLIL (Yang & Song, 2020) on the previously described ILP datasets. Times until convergence for a given edge are reported in seconds. Both FUSE-ILP and NLIL perform well on the recursive chain-like reasoning task Even-Successor. FUSE-ILP also demonstrates that it can quickly learn all chain-like definitions for the Kinships task, while NLIL (Yang & Song, 2020) failed to converge to a solution. FUSE-ILP was able to obtain an F1-score of 0.8 on the community dataset, misclassifying a single pair of parents. The rule learned for the community dataset demonstrates much of the ground truth rule structure.

$$\forall X, Y (\text{ParentalFriend}(X, Y) \leftarrow \exists Z_1, Z_2, Z_3 (\text{Parent}(X, Z_1), \text{Parent}(Y, Z_2),$$
$$\text{HasMet}(Z_1, Y), \text{HasMet}(Z_2, X),$$
$$\text{HasMet}(Z_2, Z_3), \text{ChildhoodFriend}(Z_1, Z_2))).$$

This rule is close to the ground truth, however it excludes the mutual school or football club attendance edges between children. Nonetheless, this example demonstrates that it is possible for neural ILP techniques to learn non-chain-like rules which chain-like methods can't.

## 6 CONCLUSION

In this paper we introduce FUSE-ILP, a technique that enables differentiable learning of subgraph-like rules first-order logic rules. Using tree-like meta rules and soft variable pruning and merging, we demonstrate that FUSE-ILP can solve tasks that chain-based backwards chaining neural ILP techniques fail at. We also introduce a simple, yet challenging dataset that exposes the limitations of current differentiable ILP techniques. Future work will evaluate how this technique extends to large and complex real-world datasets.

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

# A    APPENDIX

## A.1    DIFFERENTIABLE SET OPERATIONS

Previous sections have framed the inference algorithm in terms of discrete set operations. This section will describe how these discrete operations are relaxed into continuous and differentiable approximations of set operations.

During training, tuples of related entities and the subgraph regions surrounding them, $(x, y, \mathcal{G}) \in \mathbb{D}$, are drawn from a dataset. The entities and graph are both represented numerically as tensors. Entities $x, y$ are encoded into 1-hot column vectors $\boldsymbol{x}, \boldsymbol{y}$, representing an index on the entities in $\mathcal{G}$. All SAT sets $\mathbb{Z}_k$ are likewise represented as column vectors $\boldsymbol{z}_k$, where each entry is bounded $[0, 1]$.

The graph itself, $\mathcal{G}$, is encoded into a tensor of 1-hot adjacency matrices, $\mathbf{A}$, such that each entry $\mathbf{A}_{i,j,k}$ represents the existence of an edge of type $k$ from entity $j$ to entity $i$.

Under this formulation, set intersection is differentiably approximated as an element-wise minimum of two vectors.

$$\mathbb{Z} \cap \mathbb{Z}' \approx \min(\boldsymbol{z}, \boldsymbol{z}') \tag{15}$$

The parent and child operators $\phi_p(\psi), \phi_{p^{-1}}(\psi)$ described in equation 1 and equation 2 are numerically implemented as a matrix multiplication between a vector $\boldsymbol{z}$ representing the argument SAT set[6], and the adjacency matrix $\mathbf{A}_{:,:,p}$ for the predicate edge type $p$.

$$\psi_k' = \phi_p(\psi_k) \approx \mathbf{A}_{:,:,p} \boldsymbol{z}_k \tag{16}$$

$$\psi_k' = \phi_{p^{-1}}(\psi_k) \approx (\mathbf{A}_{:,:,p})^T \boldsymbol{z}_k \tag{17}$$

Generally, $\mathbf{A}_{:,:,p}, \boldsymbol{z}_k$ are not guaranteed to be stochastic, so each updated message $\psi_k'$ is clipped to remain in the range $[0, 1]$.

To improve the expressivity of rules learned by FUSE-ILP, the direction of an edge in $\mathcal{G}$ is divorced from the direction of message passing inference in $\mathcal{R}$. This is accomplished by re-defining $\mathbf{A}$ to be the concatenation of $\mathbf{A}$ and $\mathbf{A}^T$ along the predicate dimension.

$$\mathbf{A} := [\mathbf{A}_{:,:,1}, ..., \mathbf{A}_{:,:,P}, \mathbf{A}_{:,:,1}^T, ..., \mathbf{A}_{:,:,P}^T] \tag{18}$$

This allows each factor in the rule graph to represent an arbitrary ordering of arguments in its head. For instance, $\mathbf{A}_{:,:,1}$ models the relationship $P_1(Z_{\text{parent}}, Z_{\text{child}})$, while $\mathbf{A}_{:,:,1}^T$ models the relationship $P_1(Z_{\text{child}}, Z_{\text{parent}})$.

As with previous differentiable logic programming techniques, (Yang et al., 2017; Yang & Song, 2020), FUSE-ILP relaxes these operations further, defining a discrete probability distribution over each predicate's associated factor function and computing expected messages on these distributions during rule inference.

## A.2 DIFFERENTIABLE PREDICATE SELECTION

During training, the choice of factor function in each factor of the rule factor graph is relaxed, and the expected message is computed with respect to a parameterized distribution over possible predicate choices, $f_p(p; \theta)$.

$$\mathbb{E}_p[\psi_k'] = \mathbb{E}_p[\phi_p(\psi_k)] \approx \sum_{p \in \mathbb{P}} f_p(p; \theta) \mathbf{A}_{:,:,p} \boldsymbol{z}_k \tag{19}$$

This expectation can be simplified by calculating the expected transition matrix $\tilde{\mathbf{A}}$, and then performing the matrix vector product.

$$\tilde{\mathbf{A}} = \sum_{p \in \mathbb{P}} f_p(p; \theta) \mathbf{A}_{:,:,p} \tag{20}$$

Each factor in the rule factor graph maintains its own $f_p(p; \theta)$. These distributions can be parameterized arbitrarily, and this paper investigates two possible methods constructed using deep neural networks.

## A.3 DIFFERENTIABLE PRUNING

Every leaf variable in the rule tree has a probability mass associated with its membership in the "prune" group. This value can be found at the variable's pruning index in the expected group partition matrix. This value is interpreted as the probability that a leaf will be pruned from the tree. During message passing, the leaf variable's SAT vector is weighted by this probability and summed with a vector of 1s weighted by the complement probability. This operation performs a soft removal of that variable's constraints from the message passing process.

---

[6]Note that messages $\psi$ are just SAT sets like $\mathbb{Z}$. The $\psi$ notation is adopted for consistency with typical message passing formulations.

Table 2: Group Partition Matrix

|       | $g_1$ | $g_2$ | $\perp$ |
|-------|-------|-------|---------|
| $Z_1$ | 1     | 0     | 0       |
| $Z_2$ | 0     | 1     | 0       |
| $X$   | 0     | 0     | 1       |
| $Y$   | 1     | 0     | 0       |

Similarly, each splitting variable in the tree receives a pruning probability for the sub-trees branching off of it. These pruning probabilities are the minima of the pruning probabilities in each sub-tree. If a sub-tree has a high pruning probability, it it softly ignored during message passing. Using the sub-tree minima ensures that a branch is not eliminated until all of its descendant variables have been pruned from the tree.

### A.4 DIFFERENTIABLE LEAF PARTITIONING

Table 2 illustrates the group partition matrix for the hypothetical variable grouping scheme wherein $Z_1$ is merged with $Y$, $Z_2$ remains unchanged, and $X$ is unassigned to any leaf variables.

### A.5 COMMUNITY DATASET GROUND-TRUTH

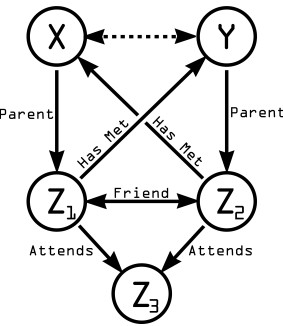

Figure 3: Ground truth rule for friendships between parents of different children in the community dataset.

Figure 3 illustrates the ground-truth rule for classifying whether two parents might become friends. The dataset is crafted such that only pairs of parents whose children satisfy all the relationships in Figure 3 will be potential friends. Critically, this rule cannot be expressed as a conjunction of independent chains. It can, however be captured by FUSE-ILP's subgraph-type rule learning.

