# OpenReview forum: "Efficient Subgraph Rule Induction via Tree Folding in Differentiable Logic Programming"
_ICLR.cc/2024/Conference — Submitted to ICLR 2024_

### Official Review · Reviewer_qyFE · 2023-10-30

**Soundness:** 2 fair
**Presentation:** 1 poor
**Contribution:** 1 poor
**Rating:** 3
**Confidence:** 4

**Summary:**

This paper tackles improving the scalability of the problem of differentiable inductive logic programming, a task of
learning suitable logic programs from data. The authors say that previous ILP algorithms
have scalability issues. The paper proposes some techniques to improve scalability.
The paper experimentally evaluates the performance by comparing it with a baseline method.

**Strengths:**

The scalability issues of differentiable ILP is an important topic. It prevents the method from being used in broader situations.

**Weaknesses:**

**The paper is very hard to follow.**
This paper is tough to follow. I have to confess that I cannot understand what is the key difference of the proposed
method compared with existing differentiable ILP methods.
This is mainly because the paper does not provide many definitions and background knowledge needed to understand the paper.

For example, there is no explanation of the differentiable ILP task. Also, the task's input and output seem not explained.
Moreover, there is no definition of first-order logic used in this paper.
The background section starts with explaining multi-hop reasoning as a problem on graphs. How are these graphs related to ILP?
The paper must clearly show how differentiable ILP relates to graph problems.

What are the tree-like structures? I think the paper should give a formal definition of tree-like structures.
What are the messages passed among graphs? Is a message a real-valued vector?

In summary, the paper might contain important ideas that might be useful for the ML community. However, the paper is almost
impossible to understand in its current form for many readers. Therefore, I suggest a major revision to improve its presentation.


**Results of experimental evaluations are weak:**
In experiments, the paper compares the proposed method with a baseline. This section is weak because:
- There are no explanations of the details of experimental settings.
- The proposed method is compared with only one baseline method, and the reason why the paper compared with NLIL is unclear. I think the paper should compare with more baseline differentiable ILP methods.
- Experimental results only report time and F1 scores. They seem insufficient to judge that the claim of the paper is correct.

**Questions:**

None

---

> ### Author Response · Authors · 2023-11-23
>
> Thank you for your detailed and well reasoned feedback, we appreciate the time that you spent reviewing our work.
>
> To address some of the questions that you raised:
> We have expanded the introduction and motivation of the technique to illustrate why subgraph rule learning is of significance to real world problems. We have also reorganized and clarified the discussion and presentation of the technique, including better high-level descriptions and a new figure to provide improved intuition for the reader. We have improved our explanation of the problem formulation and learning task, as well as the definitions of many of the domain concepts. We have removed the discussion of multi-hop reasoning as it seemed to muddle the presentation of our method. We provide a more clear explanation of how our method formulates ILP as a graph link prediction problem. We have expanded the introduction of our experiments to provide more context on why they were selected and what characteristics of the algorithms they test. We have provided a clarified explanation of the motivation for selecting our baseline method.
>
> Again, we appreciate your time and high-quality feedback.

---

### Official Review · Reviewer_Y7bE · 2023-10-30

**Soundness:** 3 good
**Presentation:** 2 fair
**Contribution:** 2 fair
**Rating:** 5
**Confidence:** 3

**Summary:**

A new approach for performing ILP using neural network parameterization is proposed. The main contribution is an approach capable of learning a broader set of rules than what is currently possible with state-of-the-art neural ILP methods. The proposed method relies on evaluating/verifying SAT on tree-like FOL structures by adapting a factor-graph message passing approach similar to existing methods. The novel aspect is “folding” which is to use this to learn more complex FOL structures by introducing constraints for merging similar structures.

Further, to learn using neural methods, each of the discrete operations in the evaluation and learning of FOL structures is encoded as a differentiable operation on a continuous tensor representation (similar to TensorLog). Experiments are performed on 3 datasets (one synthetic and 2 other standard ones) and comparisons with an existing state-of-the-art neural ILP method show promising results.

**Strengths:**

- Learning more complex structures from neural ILPs seems like a significant contribution
- The idea of using constraints for merging to form complex structures from trees and encoding them with neural nets seems interesting

**Weaknesses:**

- The experiments dot not adequately show the impact of the proposed approach. Specifically, there is a single synthetic example (community) on which the complex rule learning outcome is demonstrated. It seems like the other compared approach fails here. There are 2 other benchmarks, but it seems like the proposed approach is not necessary here.
- The paper leverages existing approaches (e.g. TensorLog, message-passing, etc.) so it was hard to understand the novel contributions of the paper.

**Questions:**

Can there be a more comprehensive evaluation done to show that i) complex rules are required for real-world cases and ii) existing methods fail for such cases while the proposed method can effectively learn such rules.

One of the aspects shown in the experiments is also learning time. How do more complex structures affect this?

If the novel contributions were better highlighted it would be useful to evaluate significance of the proposed method.

---

> ### Author Response · Authors · 2023-11-23
>
> We appreciate your time and the valuable feedback that you have provided on our work.
>
> To address some of your questions:
> We have updated the paper to include an expanded discussion of the significance of non-chain-like relationships and we provide a motivating example to illustrate how they can appear. We have also reorganized and clarified our discussion of the mechanisms that we employ to enable subgraph-like rule learning, as well as how those differ from chain-like rule learners.
>
> Again, thank you for contributing your time to our paper.

---

### Official Review · Reviewer_x87R · 2023-10-30

**Soundness:** 2 fair
**Presentation:** 2 fair
**Contribution:** 2 fair
**Rating:** 3
**Confidence:** 4

**Summary:**

This work describes an improvement over the ground breaking work of TensorLog; The work addresses the chain nature of TensorLog rules by enabling disjunctions. Other complex graphs are essentially transformed to trees.

Unfortunately, the experimental work is weak. It largely remains to be seen whether the author's claiims are valid.

**Strengths:**

The extension proposed is interesting, and seems to have bee.n impllemented..

**Weaknesses:**

- Lack of experimental support
- As the authors mention, the message passing is an improvement on the same proposal for Tor TensorLog. Can you please further clarify your contribution in this?
- Your graph folding algorithm should be better motivtated

**Questions:**

Please, use the reference style correcttyl,*

Your graph folding algorithm seems a bit aggressive. I was hoping a bit more of motivatioan and more experiments. Did you consider comparing with TensorLog or one of the many available neurosymbolic systems,

What is the difference between T and SAT/ Thanks!

Title: you use the word efficient.  but I cold not find either theoretical or experimental suppodl

---

> ### Author Response · Authors · 2023-11-23
>
> Thank you for the feedback on our paper, we appreciate your time.
>
> To address some of the questions you had: We have expanded the introduction and motivation to better illustrate how real world problems may exhibit naturally non-chain-like behavior. We have also improved the narrative of our discussion to clarify that TensorLog is a database system that describes differentiable rule inference, and not another ILP technique. We have updated our explanations of the satisfiability derivation algorithm to improve the clarity of the SAT problem and our descriptions of it.
>
> Thank you.

---

### Official Review · Reviewer_WLDK · 2023-11-01

**Soundness:** 3 good
**Presentation:** 3 good
**Contribution:** 2 fair
**Rating:** 6
**Confidence:** 3

**Summary:**

The paper extends ILP techniques that were previously limited for checking the satisfiability of longer chains of variables to cases where the variables form tree structures or even more general cases, which are then addressed by factoring the variable graph.
The method is illustrated in a few domains, and works faster than previous algorithms, and is also able to learn rules that could not have been learned with chain-based approaches, as is demonstrated on a new artificial dataset.

**Strengths:**

This seems to be a reasonable step forward. The approach is described in sufficient detail and generality, so that an outsider can also understand the key ideas. The results show that it improves over previous work.

**Weaknesses:**

The idea of factoring the variables does not seem to be entirely new to me. I can't give a concrete reference, but I am quite certain I have seen that before, maybe in a slightly different context. In any case, the step forward does not seem to be substantial.

Also, the practical relevance is not clear to me. Apparently, the authors had to define an artificial dataset where they can show that the technique does what it is supposed to do, because many standard problems can be solved with chains.

References in the paper are weird, the authors almost always use "author (year)", also in cases where "(author year)" would be appropriate.  The first reference in the bibliography is not correctly sorted in (presumably there is something wrong in the BibTex entry).  In general, the related work is not very exhaustive, missing, e.g., recent works such as POPPER. For Metagol, no reference is given (only a pointer to a github page). It is also preferable to cite the published versions of arxiv papers, not the arxiv papers themselves.
It is interesting that the authors only provide URLs to papers by Yang et al. Why? Either provide all URLs or none.

A few typos, such as "boolean" -> "Boolean". or a comma starting a new line (2nd paragraph 4.4). A careful proof-read would certainly improve the paper (but it is generally quite readable).

**Questions:**

In the 90s, family relations were toy problems for the then state-of-the-art ILP programs. How do your domains differ from what was used then? Are these large graphs from which these relations are learned? How large? How would classical algorithms such as Foil or Aleph do on such problems?

Are there benchmark problems where trees or more general graphs are necessary, or are chains sufficient for most problems?

---

> ### Author Response · Authors · 2023-11-23
>
> We thank you for your thoughtful comments and the time taken to review our paper.
>
> To address some of your questions:
> We have added a motivating example to better illustrate how chain-like reasoning can be insufficient in important real-world domains.
> We have addressed the issues with our citation formatting and incorporated more context from relevant ILP literature.
>
> Again, we thank you for your time.

---

### Meta-Review · Area_Chair_4uGy · 2023-12-05

**Metareview:**

The reviewers all thought that there were significant issues with this submission, and the overall consensus was to reject. The author's short responses did not lead to changes in the reviewers' opinions.

**Justification For Why Not Higher Score:**

There are several issues that have not been addressed in a satisfying fashion by the authors

**Justification For Why Not Lower Score:**

N/A

---

### Decision · Program_Chairs · 2024-01-16

Reject